# Functional Role of Probiotics and Prebiotics on Skin Health and Disease

Vasiliki Lolou and Mihalis I. Panayiotidis *

Department of Applied Sciences, Northumbria University, Newcastle Upon Tyne NE1 8ST, UK;
vasiliki.lolou@northumbria.ac.uk
* Correspondence: m.panagiotidis@northumbria.ac.uk; Tel.: +44-(0)191-227-4503

**Abstract:** Scientific and commercial interest of probiotics, prebiotics and their effect on human health and disease has increased in the last decade. The aim of this review article is to evaluate the role of pro- and prebiotics on the normal function of healthy skin as well as their role in the prevention and therapy of skin disease. *Lactobacilli* and *Bifidobacterium* are the most commonly used probiotics and thought to mediate skin inflammation, treat atopic dermatitis (AD) and prevent allergic contact dermatitis (ACD). Probiotics are shown to decolonise skin pathogens (e.g., *P. aeruginosa*, *S. aureus*, *A. Vulgaris*, etc.) while kefir is also shown to support the immunity of the skin and treat skin pathogens through the production of antimicrobial substances and prebiotics. Finally, prebiotics (e.g., Fructo-oligosaccharides, galacto-oligosaccharides and konjac glucomannan hydrolysates) can contribute to the treatment of diseases including ACD, acne and photo aging primarily by enhancing the growth of probiotics.

**Keywords:** probiotics; prebiotics; skin health; skin disease; dermatitis; skin infections

## 1. Introduction

Fermented food has been part of our diet, in addition to being used for therapeutic purposes, as early as 7000 BC from Egyptians, Greeks and Italians [1–3]. Some of the most ancient fermented foods used in history is wine, bread and milk products such as yoghurt. In fact, it is documented that Georgians were using wine in their diet as early as 6000 BC, whilst fermented dairy products were used for the treatment of diarrhea and other gastroenteric infections [4,5]. The relationship between human health and microbiota was first mentioned in 1907, by Elie Metchnikoff, when the enhanced longevity due to the intentionally present bacteria in yogurt was described [6]. In addition, fermented food became famous after Werner Kollath first introduced the term "Probiotic". The food industry has used probiotics in their products as an aiding ingredient and/or as a preservative means since 1989 [7]. With the evolution of food processing and preservation and the consumer's interest for a healthier and more balanced diet, probiotics became one of the most marketable ingredients. According to the World Health Organization (WHO), probiotics are live microorganisms that "when administered in adequate amounts, confer a health benefit on the host" [8]. Most common species of probiotics belong in the families of *Lactobacillus*, *Bifidobacterium* and *Streptococcus* [9] with the first two families being mostly used in studies related to human health [10]. As these microorganisms are naturally found in the gut microbiota, most studies are focused on their effects in the context of the natural function in the gut and as preventive or therapeutic agents against disease development [11–18]. To this end, probiotics have been used for the study and treatment of intestinal diseases such as gastroenteritis [19], intestinal hyperpermeability [20], urinary tract infection [21], intestinal dysbiosis [22], irritable bowel syndrome [23], Crohn's disease [24], colon cancer [25,26], ulcerative colitis [27,28] and peptic ulcer [23]. In particular, many studies have shown their involvement in regulating signaling molecules like

NFκB, MAPK, PPARγ, HSP, etc. by either activating or inhibiting their expression profile depending on the microorganism studied. Such effect(s), in turn, can trigger other signaling events including perturbations in the (i) phosphorylation content of IκBα, (ii) activation status of p38, (iii) inhibition of nuclear binding by p65 as well as (iv) induction of PPARγ mRNA levels [29–61]. In addition, probiotics have been extensively utilized in the context of intervention studies towards prevention and/or treatment of a number of human diseases including those of the skin like atopic dermatitis [AD] [62–69], allergic rhinitis [66,70,71] and wound healing [72–79] being some of the major ones (Figure 1).

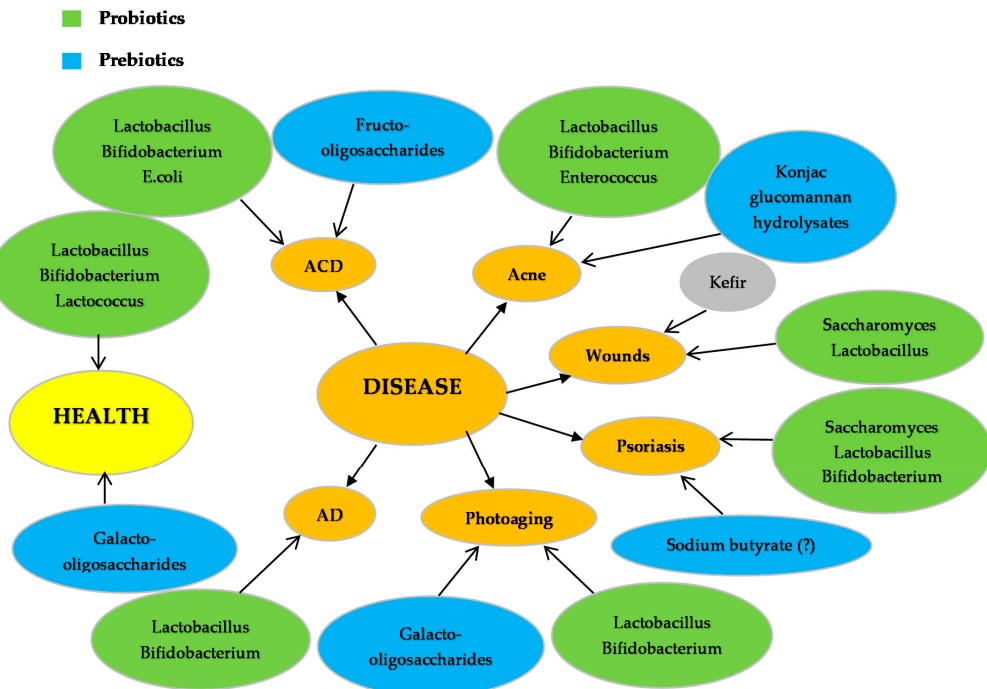

**Figure 1.** The role of probiotics and prebiotics on skin health and disease including Allergic Contact Dermatitis (ACD), Acne, Wounds, Psoriasis, Photoaging and Atopic Dermatitis (AD).

On the other hand, with the term "prebiotics" we refer to specific fermented components that enhance changes in the composition and the activity of the gut microflora in favor to the host [80]. Prebiotics are characterized by low dosage activity, absence of side effects and persistence through the gut [81]. The most commonly known prebiotics are oligosaccharides (OS; e.g., glycans), fructans (inulin-type), sugar alcohols and complex polysaccharides (e.g., β-glucans, cellulose) [82,83]. The available literature on prebiotics and their effect on human health is limited, compared to the probiotics, and it is often included in several probiotic studies. These non-digestible compounds are known for their bifidogenic effect, which varies depending on the type of prebiotic. This is based on the fact that long-chain OS are fermented in the entire gut whereas the short-chain ones are only processed in the ascending colon and the caecum. Breast milk mostly consists of prebiotic OS and as being the first food for infants; it provides the initial intestinal microbiota whose growth is supported by these OS. Furthermore, recent studies have shown the ability of prebiotics to enhance calcium absorption and have an effect on bone structure as well [82]. Moreover, these compounds are shown to affect the immune system by increasing IgA, CD4+ cells, INF-γ and IL-4 in spleen and mesenteric lymph nodes [84–86]. Additionally, other studies on healthy participants have shown a decrease of toxic fermentation metabolites in the colon (e.g., [$H_4$] tyrosine and lactose-[N]ureide) after consumption of pro- (e.g., *L. casei*) and prebiotics (e.g., n9; lactulose) [87].

Finally, the skin represents the largest organ in the human body and as such, its main function is to act as a barrier to extrinsic factors including physical, chemical and microbial threats. In this

context, a strong symbiotic relationship between microorganisms exists that constitutes its microbiota. This natural microflora supports the immune system in various ways including the production of natural antimicrobial compounds (e.g., lactic acid) as well as activation of various signaling pathways and modulation of the inflammatory response [88,89]. In this review article, we aim to focus on the beneficial role of pro- and prebiotics on skin health as well as their therapeutic and/or preventive role on specific skin diseases.

## 2. Probiotics and Prebiotics on Skin Health

There is a rather small number of studies on healthy subjects to show a beneficial effect of probiotics on skin health (Figure 1) [18,61,90–92]. In one such study, when the *L. lactis* strain; H61 was supplemented on middle-aged women, daily for eight weeks, an improvement on skin elasticity and body characteristics were observed (e.g., skin appeared more hydrated and the hair follicles had improved) [92]. Similarly, in another such study, oral intakes of *L. plantarum*; HY7714 from a group of subjects aged 41–59 years old also confirmed the effect of probiotics on increasing skin moisture, decreasing the depth of existing wrinkles and improving the overall skin gloss and elasticity [61]. Moreover, other studies have shown that when probiotic and para-probiotic *L. reuteri* were administrated orally, for 12 weeks, an increase in melanin and a decrease in Trans-Epidermal Water Loss (TEWL) were observed [91]. Such effects are in agreement with studies utilizing other probiotics (e.g., *L. rhamnosus*, *B. breve* Strain Yakult, *L. lactis*, *S. thermophilus*) and prebiotics (e.g., galacto-oligosaccharides; GOS) (Figure 1) all of which have indicated (i) improved levels of skin hydration and cathepsin-L-like activity levels (an indicator of keratinocyte differentiation and a marker of skin barrier function) as well as (ii) reduced urine and serum phenol levels (e.g., toxic by-products formed by gut bacteria) [90,93].

## 3. Probiotics and Prebiotics on Skin Disease

### 3.1. Dermatites

#### 3.1.1. Atopic Dermatitis

Atopic Dermatitis (AD), also known as atopic eczema, is a skin inflammatory disease that is observed in early stages of life and is linked with allergic rhinitis, food allergies and asthma, all of which are more prevalent in children suffering from this disease. One of the most common symptoms of eczema, apart from itchiness, is the reduction of barrier function that leads to allergen exposure and overall reduction of the TEWL, leading to dry skin [94]. In an AD model, allergens can penetrate the stratum corneum, which is altered by the epidermal epithelium deformities. Moreover, symptoms include the presence of pathogenic microorganisms, such as *S. aureus*, that colonize and infect the subjects. Another significant aspect of AD is its relationship with the gut microbiota. More specifically, the balanced microbial profile of the mucosa can promote the production of immunoglobulin A (IgA) which supports the defensive mechanisms of the gut membrane, whilst enhancing the expression of the Transforming Growth Factor (TGF) [95]. A relationship between the gut microflora and the development of AD was also observed in infants at high risk for developing AD showing an increased number of clostridia compared to control, disease free infants [96].

Specific probiotic microorganisms are shown to have a preventing role on AD and mediate the symptoms of the disease (Figure 1). They appear to do so by influencing a number of biological processes not only in AD but rather in a wide range of skin diseases (e.g., acne, psoriasis, photo aging, wounds, etc.) (Table 1 and Figure 2). More specifically, in a recent study, supplementation with *L. rhamnosus* in combination with *L. reuteri* improved the severity of eczema by 56% in children suffering from AD [65]. Moreover, in another study, *L. rhamnosus* was utilized as a supplemented probiotic, to women four weeks before delivery and six months postnatal, demonstrating to significantly reduce the risk of children developing AD during their first seven years of age [66]. Finally, when infants at high risk of developing AD were supplemented with a mix of probiotic microorganisms (e.g.,

*L. acidophilus*, *B. bifidum* and *B. lactis*), during pregnancy and after birth, they showed a reduction of immunoglobulin E (Ig-E) associated eczema by 40% [62].

**Table 1.** Probiotics and their effect on skin diseases.

| Probiotics | Disease | Function | Reference |
|---|---|---|---|
| *L. rhamnosus* | AD [1] | Improvement of severity of eczema, reduction of risk of AD development in infants | [65,66] |
| *L. reuteri* | AD<br>Infections (*S. aureus*) | Improvement of eczema. Blocks integrin, Reduces cell death due to *S. aureus* infection | [65,97] |
| *L. delbrueckii subspecies bulgaricus* | Acne | Improvement of Acne symptoms (Acne Vulgaris) | [98] |
| *L. sporogenes* | Psoriasis | Improvement of symptoms, reduction of blood sugar levels and fever | [99] |
| *L. plantarum* | Photoaging | Inhibition of MMP-1, MMP-2, MMP-9 and MMP-13 [2], enhancement of procollagen expression, inhibition of phosphorylation of Jun N-terminal kinase, increase of palmitoytransferase mRNA levels, decrease of ceramide mRNA levels, reduction of wrinkles and epidermal thickness | [100,101] |
| *L. fermentum* | Infections (wounds) | Production of gNO [3], increases productions of IL-1 [4] and TGF-$\beta$ [5] cytokines | [102,103] |
| *L. acidophilus* | AD<br>ACD [6]<br>Infections (*S. aureus*)<br>Acne | Reduction of Ig-E [7], reduction of eczema, Increase of TGF-$\beta$, Foxp3 [8], IFN-$\gamma$ [9] and IL-10 [10] expression, Inhibition of *S. aureus* infection, reduction of acne symptoms | [62,98,104, 105] |
| *L. casei*<br>*L. salivarius* | ACD<br>Infections (MRSA) [11] | Reduction of skin inflammation, inhibition of IFN-$\gamma$, CD8$^+$ T cells, increase in IL-10 production, activation of CD4$^+$CD25$^+$ T cells, inhibition of MRSA | [105–107] |
| *B. bifidum* | AD<br>Acne | Reduction of Ig-E, reduction of development of AD in infants, reduction of Acne Vulgaris symptoms | [62,98] |
| *B. lactis* | AD | Reduction of Ig-E, reduction of development of AD in infants. | [62] |
| *B. pseudolongum* | ACD | Reduction of allergic reaction on mice | [108] |
| *B. longum* | Photoaging | Prevention of TEWL [12], reduction of skin erythema, increase of mRNA expression of CD44, TIMP-1 [13] and Col1 [14]. | [109] |
| *B. breve strain Yakult* | Photoaging | Prevention of loss of elasticity, suppression of elastase, activation of IL-1$\beta$ | [38,110] |
| *B. infantis* | Psoriasis | Reduction of plasma TNF-$\alpha$ [15], increase of IL-6 | [111] |
| *S. epidermidis* | Acne | Growth inhibition of Propionibacterium acnes and Acne Vulgaris by competitive exclusion | [112] |
| *E. faecalis* | Acne | Reduction of inflammation areas, production of bacteriocins | [113] |
| *E. coli Nissle 1917* | ACD | Increase of TGF-$\beta$, Foxp3, IFN-$\gamma$ and IL-10 expression | [114] |
| Kefir grains | Infections | Production of antimicrobial substances (lactic acid, acetic acid, hydrogen peroxide, bacteriocins), Healing of *P. aeruginosa* infected wounds, Inhibition of *S. aureus*, *S. salivarius*, *S. pyogenes*, *P. aeruginosa*, *C. albicans*, *S. tympimurium*, *L. monocytogenes* and *E. coli* growth | [115,116] |

[1] Atopic Dermatitis; [2] Matrix Metalloproteinases (MMPs)-1,-2,-9,-13; [3] Nitric Oxide; [4] Interleukin 1; [5] Transforming Growth Factor $\beta$; [6] Allergic Contact Dermatitis; [7] Immunoglobulin E; [8] Forkhead box P3; [9] Interferon gamma; [10] Interleukin 10; [11] Methicilin Resistant Staphylococcus aureus; [12] Trans Epidermal Water Loss; [13] Tissue inhibitor of metalloproteinases 1; [14] Collagen 1; [15] Tumor Necrosis Factor.

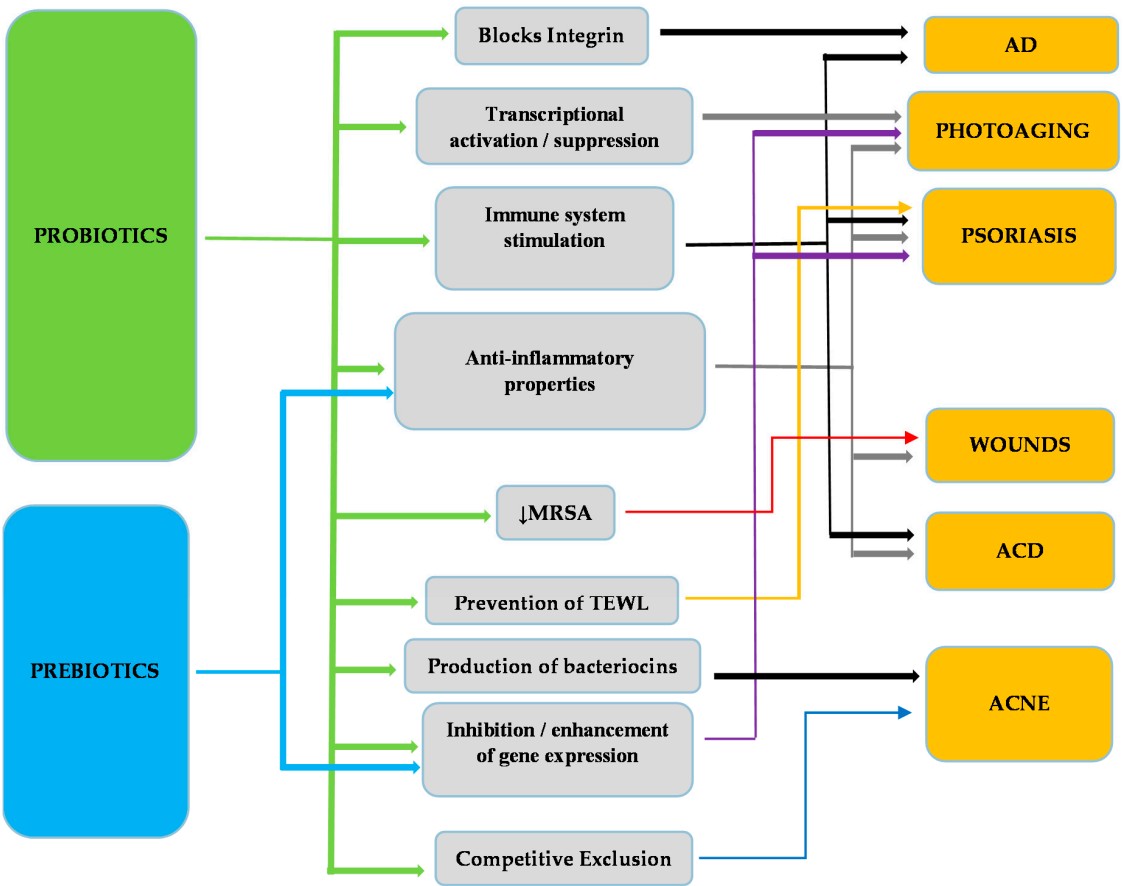

**Figure 2.** Linkage of various skin diseases with their respective mode of action through which pro- and prebiotics exert a beneficial effect. Methicilin Resistant Staphylococcus aureus (MRSA); Trans Epidermal Water Loss (TEWL).

### 3.1.2. Allergic Contact Dermatitis

Allergic contact dermatitis (ACD), also known as eczema, is caused after the skin comes in contact with an allergenic substance capable of causing an allergic reaction. Symptoms vary but include skin inflammation, itchiness, dry skin, blisters, etc. The allergic reaction is regulated by CD4$^+$ T cells in a manner where peptides derived from allergens activate Th2-type cytokines (produced by these CD4$^+$ T lymphocytes) including interleukins 4, 5 and 13 [117]. Overall, pro- and prebiotics are shown to have a preventing role on ACD and consequently mediate its symptoms (Figure 1).

*L. casei* is found to reduce skin inflammation either by targeting the inhibition of INF-$\gamma$ (responsible in producing CD8$^+$ effector T cells) [106] or via mechanisms that include the involvement of regulatory CD4$^+$ T cells [107]. In addition, the microorganism has also been shown to increase the production of IL-10 by promoting the activation of CD$^+$4CD25$^+$ Tregs thus further supporting its specific mode of action against skin inflammation [107] (Table 1 and Figure 2). On the other hand, *E. coli* Nissle 1917 (EcN) is another probiotic microorganism shown to prevent ACD by means of increasing the number of Foxp3$^+$ cells (suppress antigen priming of lymphocytes) as well as the expression of TGF-$\beta$, IFN-$\gamma$ and IL-10 (regulatory cytokine network) thus suggesting an immunomodulatory function against allergen-induced dermatitis [114] (Table 1 and Figure 2). Similar observations were made in the case of the para-probiotic *L. acidophilus* strain L-92 which was also shown to induce the activation of CD$^+$4CD25$^+$3$^+$ Tregs and consequently suppress ACD [104] (Table 1 and Figure 2).

Finally, in another study, consumption of the prebiotic fructo-oligosaccharide resulted in suppressed skin inflammation due to a favorable change in the population of the intestinal microbiota by means of increasing the population of *B. pseudolongum*. This, in turn, has led to reduced contact

hypersensitivity associated with proliferation of *B. pseudolongum* in the intestinal tract of the mice [108] (Table 2).

**Table 2.** Prebiotics and their effect on skin disease.

| Prebiotics | Disease | Function | Reference |
|---|---|---|---|
| Fructo-oligosaccharides | ACD | Reduction of allergic reaction. | [108] |
| Konjac glucomannan hydrolysates (GMH) | Acne | Inhibition of Acne Vulgaris and P. acnes, growth enhancement of lactic acid bacteria. | [118,119] |
| Galacto-oligosaccharides | Photoaging | Prevention of [1] TEWL, reduction of skin erythema, increase of mRNA expression of CD44, [2]TIMP-1 and [3]Col1. | [109] |
| Sodium Butyrate (?) | Psoriasis | Increases Fas, [4] TGF-β and p52 | [120–123] |
| Oligo-saccharides | Photoaging | Modulation of the expression of elastase-type proteases through elastin receptors | [124,125] |

[1] Trans Epidermal Water Loss; [2] Tissue inhibitor of metalloproteinases 1; [3] Collagen 1; [4] Transforming Growth Factor β.

## 3.2. Skin Infections

### 3.2.1. Wounds

Most skin infections are initiated when an opening of the skin is infected with a pathogen. Briefly, when the cohesion of the skin is disrupted (either accidentally or as an effect of a disease) it forms a wound which is characterized by torn skin or by a hematoma of the tissue. In the case of a torn tissue, there are four stages descriptive of the healing process: (i) stopping the blood flow to the damaged blood vessels (hemostasis); (ii) initiating an inflammatory response which prevents potential pathogenic microorganisms to infect the wound and maintains the microbial balance of the skin; (iii) stimulating production of growth factors causing (iv) proliferation of fibroblasts and production of extracellular matrix proteins (e.g., hyaluronan and collagen) [126]. Furthermore, these stages are characterized by the involvement of other events including generation of oxidative stress [127].

There is a great scientific interest regarding the role of skin microflora in the process of wound healing as it has been shown that the absence of microbiota can decrease the healing time [128]. On another note, wound infections occur when exogenous bacteria become dominant over the systemic and local factors of host resistance. Therefore, it is only when a balance is achieved between bacteria and host that allows for the normal processes of wound healing to proceed [129]. Over the years, scientists have turned their interest to topical application of specific probiotic microorganisms in order to evaluate their effectiveness in preventing wound inflammation as well as improving on the speed of the healing process itself. In one such study, when burn wounds were treated with *Saccharomyces cerevisiae* an overall improvement on the healing process was observed [130]. More specifically, an increase in the expression levels of collagen type 1 and transcription growth factor beta 1 (TGF-β1) were observed accompanied by improved morphological and biomechanical characteristics of the healing wounds [130].

Meticillin-resistant *Staphylococcus aureus* (MRSA) is one of the most widely known pathogens with the ability to infect wounds [131]. A number of studies have shown the capacity of specific probiotics (e.g., *L. acidophilus* and *L. casei*) to act as antibacterial agents against MRSA [105] (Table 1 and Figure 2). More specifically, the growth of the pathogen was found to be inhibited and eliminated by 99% after 24 h at 37 °C incubation [105]. Moreover, in another study, three different probiotics (e.g., *L. reuteri*, *L. rhamnosus* and *L. salivarius*) were tested against *S. aureus* infection on epidermal keratinocytes [97]. Overall, it was found that *L. reuteri* and *L. rhamnosus* (but not *L. salivarius*) reduced the ability of the pathogen to induce keratinocyte cell death. This observation was directly associated with the ability of *L. reuteri* to inhibit the adherence and invasion of the pathogen to keratinocytes while *L.*

*salivarius* did not. Furthermore, the degree of protection was greater in *L. reuteri* than *L. rhamnosus* [97] (Table 1). To conclude, given that *S. aureus* adheres with the epidermal keratinocyte cells via the α5β1 integrin, it was suggested that both of the protective probiotics reduce keratinocyte cell death by competitively excluding the pathogen from the integrin's binding sites on these skin cells [97]. Finally, antibiotic properties of probiotics have been also documented in experimental settings where wounds, infected with *S. aureus*, were treated with patches of *L. fermentum*. In these experiments, it was shown an increased wound closure concomitant with production of nitric oxide (gNO) induced by the probiotic [102] (Table 1 and Figure 2). In general, gNO is known to mediate the process of wound healing through promoting the production of IL-1, TGF-β and cytokines all of which play a major role in immune response and inflammation [103].

In addition, a number of other studies have focused on topical applications of kefir and other fermented products because of their well-known anti-microbial and healing properties. Kefir is the product of milk fermentation that contains grains characterized by specific starter cultures used in the fermentation process [132]. These grains include (i) *L. kefiri*, (ii) species of the genera *Leuconostoc*, *Lactococcus* and *Acetobacter*, (iii) lactose fermenting (e.g., *K. marxianus*) as well as (iv) non-lactose fermenting (e.g., *S. unisporus*, *S. cerevisiae* and *S. exiguous*) yeasts [132]. However, there are many more microorganisms found in Kefir grains including the species *Lactobacilli*, *Streptococci*, *Lactococci*, *Enterococci*, *Bacillus*, etc. The composition of kefir grains varies depending on their origin and the microorganisms they contain [133]. Another aspect that can change the effect and the composition of kefir is the fermentation time and conditions [134–136]. Collectively, the antimicrobial activity of kefir is the result of the composition of the product that is high in lactic acid, acetic acid, hydrogen peroxide and bacteriocins all of which can have an effect on the growth of pathogens [137] (Table 1 and Figure 1). Consequently, the complexity of the kefir grains (and kefir itself) has raised the scientific interest in the context of exploring any potential effect on the growth of existing microorganisms in the human body. To this end, when *B. bifidum* PRL2010 (a dominant microorganism in the human gut) was cultured in the presence of kefir and/or kefiran (the polysaccharide produced by kefir), it was shown that the glycans present in kefir had a beneficial role on the growth of the bacteria (perhaps due to the increased transcriptional activation of genes related to the metabolisms of glycans) [138]. Furthermore, a few studies have documented a protective effect of kefir on the wound healing process [79,115,137,139]. To this end, one of the biggest challenges in wound healing is the infection of burn wounds from the antibiotic resistant pathogen *P. aeruginosa*. As a result, this pathogen is responsible for complications on serious illnesses such as hospital acquired infections and sepsis syndromes [73–75]. Experiments on burn wounds (after contamination with *P. aeruginosa* and then treatment with kefir) showed a reduction of their size accompanied by reduced healing time when kefir was administered alone than in the co-presence of silver sulfadiazine (a common topical antibiotic used for the treatment of *P. aeruginosa* on burn wounds). Such findings highlight the potential pharmaceutical use of kefir on the treatment of burn wounds [115]. Finally, in another study, burn wounds were contaminated with 8 different pathogens (e.g., *S. aureus*, *S. salivarius*, *S. pyogenes*, *P. aeruginosa*, *C. albicans*, *S. tympimurium*, *Listeria monocytogenes* and *E. coli*) and when kefir and/or kefiran were applied to the subject's infected areas the growth of these pathogens was considerably reduced [116].

### 3.2.2. Acne

Although not many studies have been conducted on the effect of pro- and prebiotics in acne, a number of them suggest a potential preventive role of pro- and prebiotics on acne thereby mediating its symptoms (Figure 1). More specifically, in a study utilizing a mixture of probiotics (*L. acidophilus*, *B. bifidum* and *L. delbrueckii*), the side effects of minocycline administration (an antibiotic used for the treatment of *A. Vulgaris*) were reduced while still being effective in exerting a synergistic anti-inflammatory effect. These results suggest a potential use of the probiotic mixture as an alternative treatment option against *A. Vulgaris* in addition to being capable of reducing adverse side effects after chronic systemic antibiotic use [98]. Acne is enhanced in the presence of the bacterium *P. acnes*.

On the other hand, *S. epidermidis* is naturally found on skin and has been shown to antagonize *P. acnes* thus highlighting its therapeutic potential against acne [112] (Table 1 and Figure 2). In another study, the therapeutic role of *E. faecalis* SL-5 on acne was also evaluated with results demonstrating that bacteriocin (CBT SL-5; an antimicrobial compound produced by *E. faecalis*) was capable of reducing inflammation suggesting the use of *E. faecalis* as an alternative approach to acne therapy thereby avoiding the extensive use of antibiotics [113] (Table 1 and Figure 2).

Finally, despite the lack of literature on the effect of prebiotics to skin disease, konjac glucomannan hydrolysates (GMH) have also been shown to inhibit *A. Vulgaris* and *P. acnes* by stimulating the growth of probiotic microorganisms including *lactobacilli*. To this end, it is noteworthy that lactic acid bacteria show selectivity towards a mannose, a glucose substrate (found in GMH), because of the nature and accessibility of these sugars as carbon sources [118,119] (Table 2 and Figure 2).

### 3.3. Psoriasis

Psoriasis is a skin condition that causes a variety of symptoms including flaky skin (patches), itchiness and redness of the area. It is a non-contagious disease and it can affect individuals of any age [140]. There are different types of the disease including pustular psoriasis, psoriatic arthritis and plaque. Even though the literature on the effects of probiotics to skin inflammation and dermatitis is extensive, little is known on their effects on psoriasis. Nevertheless, a number of studies have been conducted on the effect of pro- and prebiotics in psoriasis suggesting a potential preventive role of their action by means of mediating the symptoms of the disease (Figure 1).

In general, studies on the role of the human epidermal microbiome in psoriasis and other skin diseases revealed that *S. epidermidis* (although a permanent member of the normal human microbiota) is second most prevalent staphylococcal species only to *S. aureus* [141]. To this end, a recent study was shown that *S. aureus* was at significantly higher levels on diseased skin as opposed to *S. epidermidis* and *P. acnes* both of which were shown to be in abundance on healthy skin thereby suggesting that psoriasis is highly associated with the microbial load of the skin [142]. To this end, another study has shown that the abundance of *S. cerevisiae* is decreased in psoriasis patients and that treatment with dimethylfumarate (DMF) successfully restored its levels, a finding of utmost importance given the well-known and beneficial immunomodulatory properties of this yeast species [143]. Moreover, extensive research indicates a strong link between potential mediators of T cell activation and the development of the disease. In particular, CD4$^+$ T cells are linked with the development of psoriatic arthritis whilst probiotics regulate T cells and reduce skin inflammation and dryness of the skin [144] (Table 1 and Figure 2). In a recent case report, the probiotic microorganism *L. sporogenes* was successfully used for the treatment of pustular psoriasis as evident by an overall improvement of the appearance of lesions and patient's general condition [99] (Table 1). A year later, Groeger et al., 2013 studied the immuno-regulatory effects of *B. infantis* in patients with ulcerative colitis, chronic fatigue syndrome and psoriasis. In the case of psoriasis, reduced plasma levels of C-reactive protein (CRP) and TNF-α were observed thus highlighting the ability of *B. infantis* to reduce systemic pro-inflammatory biomarkers and thus to act as a potential therapeutic approach in treating psoriatic disease [111] (Table 1 and Figure 2).

Sodium butyrate is produced by the gut microflora [145] and it is known for its effect on cell cycle [120], tumor growth factors (TGF-β) [121] and protease enzymes [122]. In various studies utilizing human keratinocyte (HaCaT) cells it was shown that exposure to sodium butyrate induced apoptosis by 50% through up-regulation of death receptor Fas with concomitant activation of caspases 8 and 3. In addition, increased expression levels of p52 and TGF-β were also shown suggesting the involvement of cell proliferation and terminal differentiation as well [121]. Finally, a combined treatment protocol with sodium butyrate and PD153035 (an epidermal growth factor receptor inhibitor) was shown capable of enhancing keratinocyte differentiation [123]. Collectively, data suggest that sodium butyrate can act as a potential additional approach to the management of hyperproliferative skin diseases (including psoriasis) by modulating key cellular processes like apoptosis, proliferation

and differentiation (Table 2 and Figure 2). To this end, a recent study examining the gut microbial composition in psoriatic patients revealed that a reduction of butyrate microbiota producers may have an impact on the established anti-inflammatory role of this short chain fatty acid [146] and thus explain, at least partially, its preventive role in psoriasis (among other disorders) [110]. In fact, *F. prausnitzii* (one of the most common microbial inhabitants of the large intestine) serves as an important source of butyrate which, in turn, (i) provides energy for colonocytes, (ii) reduces oxidative stress and (iii) exerts anti-inflammatory action (by triggering regulatory T cells) thereby conferring immune tolerance that goes beyond the GI tract [38,100]. Finally, another study has shown that psoriatic patients possess a substantially reduced number of *F. prausnitzii* when compared to healthy controls [101].

### 3.4. Photoaging

Skin aging is considered in the context of being either extrinsic or intrinsic. Extrinsic skin aging is caused by a number of environmental factors like UVR exposure (photo aging), smoking and life style habits (diet). In particular, photo aging is characterized by a specific phenotype that includes excessive loss of skin moisture, formation of deep and thick wrinkles, age spots, discoloration, loss of collagen and overall breakdown of the elastin network of the dermis, resulting in loss of skin elasticity [109]. To date, there are few studies investigating into the effects of probiotics/prebiotics to photo aging (Figure 1). In one such study, when hairless mice were administrated probiotic-containing fermented milk together with para-probiotic *B. breve* strain Yakult, and then subjected to UVB irradiation, it was shown an improvement in elasticity and appearance of the skin [124] together with suppression of elastase and IL-1β activity levels [125] (Table 1). These findings are in agreement with another study where administration of *L. plantarum* HY7714 to hairless mice and human epidermal fibroblasts was followed by UVB exposure and inhibition of MMPs-1,-2,-9 and -13 was recorded indicating rescued procollagen expression accompanied by inhibition of Jun N-terminal kinase phosphorylation and c-Jun expression levels. In addition, wrinkles formation and epidermal thickness were also reduced [147] (Table 1 and Figure 2). Moreover, *L. plantarum* HY7714 was shown to increase the mRNA levels of palmitoyl transferase (SPT) while reducing those of ceramide in human epidermal fibroblasts [148] (Table 1 and Figure 2). Furthermore, Galacto-oligosaccharides (GOS; one of the main prebiotics found in fermented food) were evaluated either alone or in the presence of probiotics (e.g., *B. longum*) in order to assess their effects on skin disease and inflammation. It was shown that the combination of probiotics and prebiotics prevented TEWL and reduced skin erythema whilst increasing the mRNA expression of CD44, TIMP-1 and Col1 [149] (Table 2 and Figure 2). Finally, in other studies, oligo-saccharides were also shown to prevent skin aging by modulating the expression of elastase-type proteases (through elastin receptors) [150] and/or prevent damage to the skin immune system [151].

## 4. Conclusions

Scientific and commercial interest on probiotics and prebiotics as well as their effect on human health and disease has increased in the last decade. The aim of this minireview article was to evaluate the role of pro- and prebiotics on the normal function of healthy skin as well as their role in the prevention and therapy of skin disease. Whilst a number of studies have determined the mechanisms by which some of these individual microorganisms can affect specific processes involved in the pathophysiology of skin disease, others have focused on more complex natural products (e.g., kefir) known to contain a mixture of probiotics but nevertheless also capable of exerting a potent beneficial effect. Overall, our manuscript favours the idea of the utilization of probiotics as a means of prevention and/or treatment options in skin disease. Such an alternative approach can have a huge impact in the context of therapy as it will aim to reduce the use of antibiotics and thus also reduce the side effects associated with their chronic usage. However, in order to do so, the precise mechanism of their action remains to be fully elucidated, whilst further studies need to explore their benefit in managing the outcome(s) of skin disease(s) at the clinical setting.



**Author Contributions:** Conceptualization, M.I.P.; Methodology, V.L., M.I.P.; Data Curation, V.L.; Writing-Original Draft Preparation, V.L.; Writing-Review & Editing, V.L., M.I.P.; Supervision, M.I.P.; Project Administration, V.L., M.I.P.; Funding Acquisition, M.I.P.

**Funding:** This research received no external funding.

**Acknowledgments:** The authors acknowledge financial support from Northumbria University at Newcastle, UK and specifically the Multidisciplinary Research Theme (MDRT) in Bio-economy including start-up funds (M.I.P) as well as a PhD studentship (V.L).

**Conflicts of Interest:** The authors declare no conflict of interest.

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
