# Peer review of "Functional Role of Probiotics and Prebiotics on Skin Health and Disease"

_fermentation, doi:10.3390/fermentation5020041_

Round 1

Reviewer 1 Report

In 263 line mentioned Figure 1, which is missed in the manuscript. Furthermore, the discussion should be improved by resubmitting the paper. The novelty of the work is not compelling. Also, the discussion in the chapter "3.2. Skin Infections" should be improved by extra information about prebiotics influence on skin welfare (especially in "3.4 Psoriasis" and "3.5 Photo aging". 

Author Response

We would like to thank the reviewer for his/her constructive criticism which has substantially improved our manuscript.

Reviewer comment: In 263 line mentioned Figure 1, which is missed in the manuscript.

Our response: This is now corrected. In fact, there was not supposed to be a Figure 1 but rather a Table 1 instead.  

Reviewer comment: Furthermore, the discussion should be improved by resubmitting the paper. The novelty of the work is not compelling. Also, the discussion in the chapter "3.2. Skin Infections" should be improved by extra information about prebiotics influence on skin welfare (especially in "3.4 Psoriasis" and "3.5 Photo aging". 

Our response: Text has been added throughout all relevant sections (highlighted in yellow) at per the reviewer's comment. 

We hope that we have addressed the comments of the reviewer and that our manuscript is now suitable for publication.

Reviewer 2 Report

The article written by Lolou and team entitled “Functional Role of Probiotics and Prebiotics on Skin Health and Disease” is comprehensive review article which gives an idea about the probiotics and prebiotics on skin health. Probiotic bacteria have become increasingly popular during the last two decades as a result of the continuously expanding scientific evidence pointing to their beneficial effects on human health. Although several research and review articles have already been written on probiotics and prebiotics this article is the first of its kind which selectively stores the information on skin health.

I have pointed out few comments to be corrected by the authors.

Page 1 Line 11: Change “The aim of this review article was to evaluate” to The aim of this review article is to evaluate……

Page 1 Line 21: too many keywords are included. The journal requires few up to 6 keywords.

Page 1 Line 27: early as 7000BC from Egyptians To early as 7000 BC from Egyptians

Page 1 Line 28: used in history, is wine, bread To used in history is wine, bread

Page 1 Line 27: Separate the year and BC wherever applicable.

The article is little bit short to get the review article tag but may be placed as Mini review.

The article needs to include at least a figure to make the story more understandable specially the mechanism of action of probiotics and prebiotics needs to be pictured.

I suggest the authors to write few paragraphs on mechanism of action and cross talks between prebiotics and probiotics with a clear diagram.

Author Response

We would like to thank the reviewer for his/her constructive criticism which has substantially improved our manuscript.

Reviewer comment: Page 1 Line 11: Change “The aim of this review article was to evaluate” to The aim of this review article is to evaluate……

Our response: Done 

Reviewer comment: Page 1 Line 21: too many keywords are included. The journal requires few up to 6 keywords.

Our response: Done

Reviewer comment: Page 1 Line 27: early as 7000BC from Egyptians To early as 7000 BC from Egyptians

Our response: Done

Reviewer comment: Page 1 Line 28: used in history, is wine, bread To used in history is wine, bread

Our response: Done

Reviewer comment: Page 1 Line 27: Separate the year and BC wherever applicable.

Our response: Done

Reviewer comment:The article is little bit short to get the review article tag but may be placed as Mini review.

Our response: We agree with the reviewer and we have now indicated this work product as a mini review article (sentence 1 of page 1).

Reviewer comment: The article needs to include at least a figure to make the story more understandable specially the mechanism of action of probiotics and prebiotics needs to be pictured.

Our response: We agree with the reviewer and we have now included a figure to show the role of probiotics and prebiotics on skin health and disease (Figure 1 on page 2).

Reviewer comment: I suggest the authors to write few paragraphs on mechanism of action and cross talks between prebiotics and probiotics with a clear diagram.

Our response: We agree with the reviewer and we have now written a few paragraphs throughout the text (highlighted in yellow) including an additional figure regarding the linkage of various skin diseases with their respective mode of action through which pro- and prebiotics exert a beneficial effect (Figure 2 on page 6) . 

We hope that we have addressed the comments of the reviewer and that our manuscript is now suitable for publication.

Round 2

Reviewer 2 Report

Dear authors 

Thank you for working on the revision of your manuscript. Now everything looks fine and the data are updated with substantial revisions. One thing I still want from you is the quality of pictures both Figure 1 and Figure 2 does not look like a figure. I will definitely like to see an improved version of both these figures.

Thank you

Author Response

We would like to thank the reviewer for his/her suggestion(s) which we believe greatly improve the quality of our manuscript. We have revised both our figures in the revised draft of the manuscript. We hope that our revisions are acceptable and that the manuscript can be published.